# The Association of Physical Activity Level with Micronutrient and Health Status of Austrian Bank Employees

**DOI:** 10.3390/nu15234884

**Published:** 2023-11-22

**Authors:** Markus Schauer, Mohamad Motevalli, Derrick Tanous, Susanne Mair, Martin Burtscher, Katharina Wirnitzer

**Affiliations:** 1Department of Sport Science, University of Innsbruck, 6020 Innsbruck, Austria; 2Department of Research and Development in Teacher Education, University College of Teacher Education Tyrol, 6010 Innsbruck, Austria; 3Research Center Medical Humanities, University of Innsbruck, 6020 Innsbruck, Austria; 4Department of Pediatric Oncology and Hematology, Charité—Universitätsmedizin Berlin, 13353 Berlin, Germany

**Keywords:** physical exercise, sport, sedentary lifestyle, nutrition, mineral, vitamin, occupation, coenzyme Q10, homocysteine

## Abstract

Background: Favorable health benefits of an active lifestyle have been clearly documented within the context of occupational health. However, a knowledge gap exists regarding the monitoring and comparison of micronutrient status across varying levels of physical activity (PA). This study aimed to investigate the association of PA level with micronutrient status and the associated health biomarkers among a cohort of Austrian bank employees. Methods: Using a cross-sectional design, this study involved the participation of bank employees (*n* = 123; average age: 43 years; 49% males) from the federal state of Tyrol, located in the western part of Austria. To assess PA levels and sedentary behavior, the Global Physical Activity Questionnaire (GPAQ; developed by the WHO) was administered. Accordingly, participants were categorized into three groups: low PA, moderate PA, and high PA. Participants’ blood samples were obtained to measure blood levels of micronutrients, homocysteine, and CoQ10. The values of vitamins and minerals in whole-blood were compared to sex-specific reference ranges and grouped into three categories: below, within, or exceeding the reference range. Results: The prevalence of a high PA level was 61%, while 18% of participants had a low PA level. Overweight/obesity was significantly less prevalent among participants with high PA levels (22%) compared to those with moderate (50%) and low (50%) PA levels (*p* = 0.045). No significant differences between PA levels were found for sex, age, diet type, homocysteine, or CoQ10 markers (*p* > 0.05). There was no significant PA-based difference in blood concentrations of most vitamins and minerals (*p* > 0.05), except for vitamin D (*p* = 0.001) among females, as well as selenium (*p* = 0.040) and vitamin B12 (*p* = 0.048) among males. Conclusion: The present findings offer initial insights into the link between PA behaviors, micronutrient status, and health, highlighting potential implications in occupational health and lifestyle, specifically in developing tailored approaches based on PA levels.

## 1. Introduction

A healthy lifestyle is a major contributor to health and well-being, whereas unhealthy lifestyle behaviors are linked to a range of health issues, including non-communicable diseases (NCDs) [1,2]. Physical activity (PA), involving sports and exercise, is a central component of a healthy lifestyle and is considered a therapeutic approach for various health conditions [3,4]. The World Health Organization (WHO) suggests that adults should undertake at least 150 min of PA per week distributed across a minimum of three days in order to achieve favorable health results [4]. Data from large-scale studies, however, show that a substantial proportion of adults worldwide fail to meet the recommended levels of PA, with a higher prevalence in developed countries [5,6]. While policies to cultivate healthy behaviors need to be prioritized, promoting preventive measures is crucial to improve population health and mitigate the burden of NCDs. In this regard, nutritional assessment, alongside monitoring PA behaviors, is a well-established strategy for diagnosing chronic conditions and evaluating health status [7,8,9,10].

Occupational health is key in safeguarding workers’ well-being and safety, boosting their efficiency, and nurturing a favorable workplace atmosphere [11,12]. Data show that there is an association between the sedentary nature of most professions and an elevated risk of NCDs and their associated risk factors [11]. In Austria, approximately half of the adults (44% of males and 46% of females) were found to be primarily physically inactive while working [13]. Available data consistently highlights the detrimental impact of sedentary occupational activities on health [14,15,16]. It has been reported that mortality rate increases by 2% for every hour spent seated, with a further increase (up to 8% per hour) during extended sitting periods [16]. Bank employees, as a health-vulnerable group, often spend prolonged time periods sitting at desks or facing computer monitors, participating in cognitively challenging assignments that may potentially develop unhealthy eating behaviors, resulting in further negative health outcomes [17]. Given the significant impact of the occupational environment on health and well-being, even within a relatively short lifespan, it becomes imperative to implement proactive health monitoring strategies across diverse working populations.

Minerals and vitamins play critical roles in a wide range of physiological procedures, such as energy metabolism, hormone control, neurotransmitter generation, immune response, and antioxidant protection [18,19]. While micronutrients substantially contribute to the maintenance of biological homeostasis within the human body, deficiencies in micronutrients can result in various health disorders [18,19]. Evidence indicates that diseases known to be significant contributors to morbidity and mortality (including, but not limited to, cardiovascular disorders, cancer, and diabetes) are closely linked to diet, especially nutritional deficiencies [20,21]. For instance, data indicate that both obesity and metabolic syndrome, as two risk factors for NCDs, are associated with vitamin D deficiency [22,23]. Moreover, a significant connection has been reported between cardiovascular diseases (including hypertension) and deficiencies in essential minerals such as magnesium, potassium, and calcium [24]. In addition to micronutrients, several biomarkers have been identified as health indicators associated with the risk factors of chronic diseases. Homocysteine and coenzyme Q10 (CoQ10) are two examples of health biomarkers whose activities are regulated by a range of minerals and vitamins [25,26,27,28], highlighting the significant role of micronutrients in interacting with health-related biomarkers. Hence, it is crucial to emphasize the importance of monitoring micronutrient status and addressing the associated deficiencies in order to maintain overall health and well-being.

Research evidence indicates that PA is a key determinant of health and nutritional status, influencing numerous physiological and metabolic processes within the human body [29,30]. It has been documented that PA, through its influence on metabolism, nutrient transport, and cellular signaling, may impact micronutrient requirements and utilization [31,32]. Physically active individuals have greater needs in both the quantity and quality of micronutrients due to increased losses through sweat and urine and have a greater need to defend against free radicals [31]. In this regard, a significant association has been reported between PA levels, nutritional adequacy, and dietary energy density [33]. Additionally, data show that engaging in regular PA can influence the utilization, absorption, and distribution of micronutrients within the body [34,35,36]. For instance, exercise-induced changes in metabolic rate and nutrient transport mechanisms can affect the body’s demand for certain vitamins and minerals [34]. On the other hand, micronutrient status can influence an individual’s capacity to engage in PA optimally, and deficiencies in several micronutrients are linked to reduced exercise performance, impaired muscle function, and prolonged recovery periods [37,38]. These facts collectively emphasize the bidirectional association between PA and micronutrient status.

Despite the availability of evidence regarding the health benefits of PA, there remains a knowledge gap regarding the monitoring and comparison of micronutrient status across different levels of PA, especially within the context of occupational health. Understanding the links between PA and blood concentrations of vitamins and minerals offers a pathway to the potential mechanisms underpinning the interplay between health components. The aim of the present study is to examine the association of PA levels (classified as low, moderate, and high) with micronutrient status and the associated health biomarkers, including blood levels of homocysteine and CoQ10. The rationale behind this PA classification lies in understanding how varying levels of PA might modulate micronutrient and health status. Analyzing these variations can provide valuable insights into the potential synergistic effects of PA and micronutrient status on health, which can ultimately aid in developing customized nutritional strategies aimed at optimizing health outcomes for populations employed in sedentary workplaces.

## 2. Materials and Methods

### 2.1. Study Design and Sample Recruitment

This study had a cross-sectional design. The study involved the voluntary recruitment of a group of bank employees from Austria, specifically from the Tyrol region, with ages ranging from 18 to 65 years. Participation in the present research was open to all adults who were employed by a bank in Tyrol. The research design was ethically approved by the independent Ethics Committee of the Medical University of Innsbruck (Ethikkommission Nr: 1136/2022) in accordance with established international scientific and ethical protocols, including adherence to the guidelines of Good Clinical Practice and the Helsinki Declaration. The collection of data occurred as part of the regular, voluntary employee health program offered annually by Tyrolean banks, conducted at the Biogena Diagnostics Point Tyrol throughout December 2019 and January 2020. Participants were informed about the research’s objectives, processes, and potential pros and cons, and their explicit consent was acquired before their enrollment in the study. The invitations highlighted the optional nature of involvement, accentuating that participation in the study was entirely at the discretion of the individuals. Participants were informed of their right to cease their participation in the study at any moment without having to state a reason. There was no monetary reward for engagement in the study. The study commenced with an initial group of 280 participants working full time in diverse job-specific roles; however, the conclusive evaluation encompassed 123 bank employees who effectively concluded all mandatory evaluations, including blood testing and questionnaire completion. Among those who did not complete the study, one participant identified with stomach cancer—considered a severe condition with a profound impact on nutritional status—was intentionally excluded from the final sample. Throughout the study, the privacy of personal data was guaranteed, with protocols established to anonymize and handle all accumulated data with utmost security. Rigorous ethical precautions were adhered to in order to ensure the well-being, privacy, and rights of the study participants.

### 2.2. Research Procedure and Measures

Two primary measures, including blood test and survey, were conducted in this study. At the Biogena Diagnostics Point Innsbruck, a medical specialist collected blood samples to assess participant micronutrient status, encompassing mineral, vitamin, homocysteine, and CoQ10 serum levels. GanzImmun conducted the blood analysis following standardized laboratory assessment protocols [39]. In parallel, a multi-module questionnaire was administrated to assess participant physical activity (PA) levels and sedentary habits, alongside the collection of additional information, including sociodemographic factors, anthropometric characteristics, work-related details, medical history, and diet-type preference. Participants completed the printed version of the questionnaire within 20 min in the laboratory.

The questionnaire included clear instructions outlining the correct procedures for measuring and reporting body weight and height values, emphasizing the importance of precise reporting. Body Mass Index (BMI; kg/m^2^) was determined through self-reported body weight and height measurements. Following the BMI categorizations for adults established by the World Health Organization [40], participants were grouped into four BMI-based subgroups: underweight (<18.5 kg/m^2^), normal weight (18.5–24.9 kg/m^2^), overweight (25–29.9 kg/m^2^), and obese (≥30.0 kg/m^2^). In addition, participants were classified into two distinct diet-type groups: 1. those with an omnivorous or mixed diet (without particular dietary constraints), and 2. individuals adhering to a vegetarian diet (excluding meat and fish-based products) or a vegan diet (excluding any food or ingredient derived from animals) [41,42].

### 2.3. Physical Activity and Sedentary Behavior

The Global Physical Activity Questionnaire (GPAQ) was used to evaluate the participants’ PA levels and sedentary time. The GPAQ is a validated and widely used survey tool developed by the World Health Organization (WHO) to collect comprehensive data on PA and sedentary behavior in adult populations [43]. It is designed to gather information about an individual’s participation in various forms of PA and sedentary behaviors, and the responses to the GPAQ can be used to categorize individuals into different PA levels.

Participants were asked to complete the GPAQ questionnaire, providing information about their PA over a normal week. The questionnaire consists of four domains: work-related activities, transport-related activities, recreational activities, and sitting habits. The questionnaire enabled participants to report their “sedentary time”, measured in hours per week, which included the time spent sitting or reclining during a typical day, excluding sleep. For the analysis of “PA levels”, the GPAQ allowed for participants to report the duration and frequency of activities classified as vigorous-intensity, moderate-intensity, and walking. In accordance with its designated formula, PA levels were computed, and participants were categorized as having low PA, moderate PA, or high PA in accordance with the weekly PA guidelines established by the WHO [43,44].

### 2.4. Blood Sample Analysis

The blood sampling protocol was performed based on standard laboratory procedures to safeguard the integrity of the samples. After a fasting period of 12 h, venous blood samples were collected between 8:00 and 11:00 using anticoagulant tubes. Each blood sample was divided into two portions, with one earmarked for serum isolation. Following centrifugation (3 min at room temperature at 3000 rotations per minute), the serum was separated from the red blood cells, carefully transferred into cryovials, and promptly sent to the GanzImmun laboratory for comprehensive analysis.

Analyses of micronutrient concentrations, including potassium, calcium, magnesium, copper, iron, zinc, selenium, manganese, molybdenum, vitamin B6, vitamin B9 (folate), vitamin B12, and vitamin D, were conducted using the whole-blood samples. These assessments followed established laboratory protocols [39]. Given the predominant distribution of many micronutrients within blood cells, a thorough examination of whole-blood content offers more profound insights compared to serum measurements [39,45,46]. Evidence shows that the fundamental biochemical activities of minerals and trace elements predominantly transpire within cellular structures [47], which highlights that serum values may not accurately indicate cellular concentrations. In light of this, it seems that the examination of blood cells, which mainly arise from metabolically dynamic bone marrow, offers a more exact means to assess the metabolic state of these micronutrients. As a result, a comprehensive analysis of blood cells was performed in this study to ensure a more precise reflection of micronutrient status.

The levels of serum homocysteine were determined using chemiluminescence immunoassay (CLIA) kits with a detection range for reagent from 1 to 50 µmol/L. A senior laboratory technician conducted the analysis using a Centaur XPT analyzer. For homocysteine samples, the intra-assay coefficient of variation was 3.9%, while the inter-assay coefficient of variation was 5.8%. Participants were classified into three groups based on their homocysteine levels: under 10 µmol/L, between 10 and 15 µmol/L, and above 15 µmol/L [48].

High-performance liquid chromatography (HPLC) utilizing ultraviolet detection was used to measure serum CoQ10 concentrations, assuring both accuracy and sensitivity in assessment. The CoQ10 measurements derived from the blood samples of participants were modified according to their cholesterol levels. A significant association exists between cholesterol metabolism and CoQ10, indicating the engagement of CoQ10 in the electron transport chain and its function in the synthesis of cholesterol [49]. Therefore, considering individual variations in cholesterol metabolism, which may impact the circulating levels of CoQ10 [49], adjusting CoQ10 levels based on cholesterol levels was key to enhancing the validity of measures. To perform the adjustment, the CoQ10 values were divided by the cholesterol levels to compute a CoQ10/cholesterol ratio.

### 2.5. Statistical Analysis

The statistical software R version 4.1.1 (R Foundation for Statistical Computing, Vienna, Austria) was used to conduct all statistical analyses. Descriptive statistics included mean values and standard deviation (SD) or median, range, and interquartile range (IQR) for exploratory analysis. Pearson chi-square tests (χ^2^) were applied for nominal scale data to examine the association between PA levels and sex (females and males), diet type (omnivores and vegetarians/vegans), BMI levels, and homocysteine levels. Kruskal–Wallis tests (using t or F distributions, ordinary least squares, and standard errors with R^2^) were used for ordinal and metric scale variables to assess the association of PA levels with age, body weight, height, BMI, homocysteine, CoQ10, adjusted CoQ10, and sedentary time. Box plots were constructed, including 95% confidence intervals, to visually represent potential variations in blood parameters based on PA levels. In addition, Likert plots were designed to illustrate the extent of variation in micronutrient values across the three PA levels based on the sex-specific reference ranges [39]. The statistical significance level was established at *p* ≤ 0.05.

## 3. Results

The data analysis was carried out on a final sample of 123 adults (62 females and 61 males; with a median age of 43 years) who successfully provided valid and reliable data. Regarding PA levels, 61% of participants (*n* = 75) demonstrated a high level of PA, whereas 21% and 18% exhibited moderate and low PA levels, respectively. The prevalence of overweight/obesity was 33%, while 6% of participants were classified as underweight. A significant portion of the participants (93%) indicated adhering to an omnivorous or mixed diet, and the prevalence of vegetarian/vegan diet was 7%. There was a significant difference (*p* < 0.001) in average BMI values between subgroups: low PA (25.1 kg/m^2^), moderate PA (24.9 kg/m^2^), and high PA (22.7 kg/m^2^). The analysis of BMI categories showed a significant difference between the study groups (*p* = 0.045), where participants with a high PA level showed an increased likelihood of being in normal-weight or underweight subgroups, while displaying a diminished likelihood of belonging to the overweight and obese categories. A significant between-group difference was found in sedentary time (*p* = 0.047), indicating a steady decrease in weekly sedentary time as PA level advances. No significant difference between the study groups was found for age, sex, diet type, homocysteine values, homocysteine categories, absolute CoQ10 values, or adjusted CoQ10 values (*p* > 0.05). Table 1 shows a summary of participant data based on the three categories of PA levels (low, moderate, and high).

Among female participants, a significant difference was observed between PA levels and blood concentrations of vitamin D, indicating a gradual increase in vitamin D levels (from 63.1 ± 24.3 to 101.9 ± 38.1 nmol/L) as PA level advances from low to high (*p* = 0.001). Among females, no significant difference between PA levels was found in blood concentrations of other micronutrients (*p* > 0.05). Among male participants, a significant difference between PA levels was detected in blood concentrations of selenium (*p* = 0.040) and vitamin B_12_ (*p* = 0.048). No significant difference was observed between PA levels and the blood concentrations of other micronutrients (*p* > 0.05) in male participants. Table 2 and Table 3 represent the micronutrient status of females and males across different PA levels. Figure 1 displays the differences between PA-level and sex categories (with 95%-CI) for each of the blood variables.

Figure 2 demonstrates the extent of variation in micronutrient values from the sex-specific reference norms, differentiated by PA levels. The status of vitamin D shows that a large portion of female participants in all groups, but more especially those with low and moderate PA levels, exhibited vitamin D levels below the standard range. Consistently, a remarkable deficiency in vitamin D was observed among male participants, primarily among those with moderate and low PA levels. Regardless of vitamin D levels, most males and females displayed a normal blood micronutrient status, predominantly falling within the standard reference range.

## 4. Discussion

The aim of the present study was to examine differences in micronutrient status and health parameters among a cohort of 123 Tyrolean bank employees categorized into three groups based on their PA level: low, moderate, and high. A summary of the key results is as follows: (i) the prevalence of a high PA level was 61%, while 18% of participants had a low PA level; (ii) the prevalence of excess body weight (BMI ≥ 25 kg/m^2^) was 33%, with a significant decreased rate among participants with a high PA level (22%) compared to those with moderate (50%) or low (50%) PA levels; (iii) weekly sedentary time decreased significantly as PA levels advanced; (iv) there was no significant difference between PA groups for sex, age, diet type, homocysteine, or CoQ10 markers; (v) most participants displayed a normal blood micronutrient status, primarily falling within the standard reference range; however, the prevalence of vitamin D deficiency was high in both genders, especially among those with moderate or low PA levels; (vi) among female participants, a significant increase in vitamin D levels was found as PA level advances from low to high, but no significant between-group difference was found in blood concentrations of other micronutrients (including calcium, magnesium, potassium, zinc, iron, copper, manganese, selenium, molybdenum, vitamin B6, vitamin B9, or vitamin B12); and (vii) in the sample of males, except for selenium and vitamin B12, no significant association was observed between PA level and micronutrient concentrations.

A growing body of research has indicated the significant health risks associated with sedentariness and low PA level [50,51,52]. In the present study, a majority of bank employees (constituting 61%) had a high level of PA, while 21% and 18% had a moderate and low PA level, respectively. In comparison with data from both national and international references for adults [53], Tyrolean bank employees demonstrate significantly higher levels of PA than individuals of similar age groups. Findings from a comprehensive study conducted by the Robert Koch Institute [54] indicate that the majority of German adults (~80%) do not meet the weekly PA levels recommended by the WHO, which is inconsistent with the present findings among Austrian bank employees. Results from a study on Brazilian bank employees show that around half of participants engaged in regular PA [55]. These findings may suggest that bank employees exhibit a higher-than-average rate of PA, which could be attributed to profession-related factors influencing PA levels, such as access to recreational spaces and/or corporate participation in health initiatives. In addition to these factors, it is important to acknowledge that differences in the methods used to measure PA, the potential bias associated with self-reported data, and the influence of confounding (including sociocultural and environmental) factors may also contribute to the variations in PA habits.

Data from the 2019 national report [56] indicate that approximately 51% of Austrian adults carry excess body weight in the form of overweight or obesity. In contrast, the present study found a lower prevalence, with only 33% of bank employees being overweight or obese. The difference in occupational activity levels, the availability of health-promotion initiatives, and individual lifestyle choices might be potential factors contributing to the aforementioned difference between the present finding and Austrian references. However, based on studies in non-European nations, over half of bank employees were affected by overweight or obesity [57,58,59]. This inconsistency highlights the significance of genetic, socio-cultural, and behavioral factors that can influence body composition and adiposity in varying ways among different populations [60,61,62]. When comparing the study groups, a significant difference in the prevalence of overweight/obesity was evident based on PA levels, where bank employees with low or moderate levels of PA were more than twofold as likely to have excess body weight (50%) compared their colleagues with a high PA level (22%). In this regard, it is well-established that age and sex are two critical indicators of PA levels and the associated changes in body composition [56,63,64]. While the distribution of sex and age within the present study groups was normal and insignificant, other factors such as dietary habits, stress levels, and workplace culture may play a key role in explaining the above difference. Consistently, data from large-scale Austrian studies spanning various age groups demonstrate that PA patterns are linked to the interplay of several lifestyle factors [65,66,67,68], highlighting the significance of recognizing diverse health determinants. 

Bank workers, as a professional cohort, often spend extended periods in seated positions while engaged in mentally demanding tasks, a situation that is linked to an elevated risk of all-cause mortality [16]. In the present study, an expected significant reduction was observed in weekly sedentary time as PA levels increased from low (53.4 h/w) to moderate (42.4 h/w) and high (39.3 h/w). The general sedentary time among bank employees (42.5 h/w or 364 m/day) is less than what was found in a comparable study (552 m/day) involving German adults [69]. However, sedentary behavior appears to be common among bank employees, with findings from two distinct investigations showing a prevalence of physical inactivity at 60% [70] and 82% [71]. While several intrinsic and extrinsic factors may contribute to sedentary behavior, findings from a study examining barriers to adopting healthier lifestyle habits underline that the extent of time allocated for physical exercise is a pivotal factor driving the rising prevalence of physical inactivity [72]. According to the results of a meta-analysis, prolonged periods of sedentary behavior are associated with an elevated risk of all-cause mortality, but regular engagement in PA has the potential to mitigate this risk [73]. These findings emphasize the importance of addressing sedentary behavior and promoting PA in the context of occupational health to enhance overall health status.

The present study also examined the micronutrient status of bank employees to gain deeper insights into the blood levels of mineral and vitamin concentrations across different PA levels. As documented by the DGE Nutrition Report [74] and the Austrian Nutrition Reports of 2012 and 2017 [20,21], adult populations often struggle to attain adequate absorption of numerous micronutrients (including vitamin A, vitamin D, vitamin B9, vitamin B12, magnesium, potassium, calcium, and iodine) through their typical diets, and this may be linked to various nutritional deficiencies and associated health issues. Among the present bank employees, there was no significant difference in the blood concentrations of most micronutrients between PA groups, except for vitamin D (among females) or selenium and vitamin B12 (among males), which showed higher concentrations among those with high levels of PA. Consistent with the present findings, results from a study revealed that PA level can impact nutritional adequacy, with varying effects observed between males and females [33]. The elevated vitamin D levels observed in physically active females may be partially attributed to their involvement in outdoor PA during different seasons, a prevalent practice among Austrian women [75] that exposes them to natural vitamin D absorption through sunlight exposure. Although the observed findings regarding the association between micronutrient status and PA level may be influenced by a complex interplay of physiological and dietary aspects (as well as methodological and sample-related factors), more in-depth investigations are necessary to thoroughly investigate these dynamics. When compared to sex-specific norms, the majority of participants exhibited a typical blood micronutrient status, mainly falling within the standard reference range. Nevertheless, vitamin D deficiency was notably prevalent among both males and females, particularly among individuals with moderate and low levels of PA. The elevated prevalence of vitamin D deficiency among bank employees aligns with results obtained in a comparable Austrian study on adult populations [76] and several international research findings [77,78,79]. These data highlight the widespread nature of vitamin D deficiency and emphasize the necessity for tailored strategies to address this concern. However, it is worth noting that the accuracy of the cutoff values used to categorize vitamin D deficiency among diverse population groups is a subject to consider [76,80]. In general, however, research indicates that micronutrient deficiencies are frequently observed (though with some variability) in individuals with low levels of PA [81] and also in those with high PA levels, including athletes [82]. Athletes and highly active individuals often have a greater micronutrient intake than untrained and less-active people [31,83]; however, there is no agreement on whether individuals with high PA levels should have greater micronutrient needs when compared to less-active populations [84,85]. Collectively, these findings highlight the importance of evaluating nutrient status and emphasize the need for tailored approaches to address micronutrient deficiencies, with special attention directed toward vitamin D, which stands out as the primary concern across all PA levels.

As part of the present study, the participants’ blood levels of homocysteine and CoQ10 were also examined to further the understanding of the association of these two health biomarkers with PA levels. While no significant differences were observed in homocysteine values, homocysteine categories, absolute CoQ10 values, and adjusted CoQ10 values between PA groups, it is worth noting that 9% of the total participants had a blood homocysteine level exceeding 15 μmol/L, a condition known as hyperhomocysteinemia [86]. This finding aligns with data from a similar study, where the prevalence of hyperhomocysteinemia among German adults was reported to be 7% [87]. Inconsistent with the present study, existing evidence indicates that physically active adults had lower concentrations of homocysteine when compared to their less-active counterparts [88,89]. However, this particular association has not been observed among children and adolescents, as indicated by other research [90]. Nevertheless, in line with another aspect of the current study, findings from a study indicate that there was no significant association between PA level and the absolute or cholesterol-adjusted values of CoQ10 among adults [91]. However, when analyzing these factors among young and older adults separately, a significant association emerges. Among older adults, there was an increase in CoQ10 concentration as PA levels increase, while among younger adults, there was a decrease as PA levels increase [91]. These findings collectively unfold the complexity of the relationship between PA levels and health biomarkers such as homocysteine and CoQ10, which may be influenced by sociodemographic factors, particularly age. This outcome suggests a pressing need for in-depth investigations into the various factors contributing to this intricate association within the context of occupational health and wellness. In this regard, it should be considered that the biological condition of different professional groups can be influenced by both innate biological predispositions and the long-term impact of the working environment [92].

This study has some limitations that require acknowledgment. Firstly, the cross-sectional design limited the examination of cause-and-effect associations between the study variables and health outcomes. Moreover, despite our efforts to collect accurate data (particularly providing clear measuring and reporting explanations), relying on self-reported body weight and height measurements may compromise the precision of BMI values, and subsequently, impact the accuracy of BMI classifications. In addition, relying on self-reported data about PA, even though the PA questionnaire was validated, raises concerns about potential recall and reporting biases, which could have resulted in the underreporting or overreporting of PA behaviors. Another limitation is the study’s exclusive focus on bank employees, which may limit the generalizability of the findings to other occupational sectors and the general adult population, given the fact that employment status is considered a predictor of health [93]. Additionally, confounding factors were not taken into account during the data analysis, including participants’ eating habits, supplement intake, socioeconomic status, as well as job-specific roles. In this regard, it is important to note that there could be sex- or age-based differences across the study groups, which should be considered a limitation. Despite the above-mentioned limitations, this study’s notable strength lies in its thorough evaluation of various health parameters. This approach contributes to a well-rounded comprehension of the health status of Austrian bank employees. The findings have significant implications for public health campaigns and workplace initiatives, particularly in the domain of occupational health. The examination of PA levels in the study highlights the significance of encouraging an active lifestyle among bank employees. In this regard, interventions implemented in the workplace, such as organized exercise initiatives, have the potential to promote higher levels of PA and reduce sedentary behavior. Additionally, routine health assessments and educational workshops aimed at promoting well-being could equip bank employees with practical health knowledge and guidance. These efforts might improve awareness regarding issues like healthy lifestyle, nutrient deficiencies, and the associated chronic diseases, ultimately fostering a culture of proactive health management.

In future research efforts, longitudinal approaches can be applied to investigate the temporal relationships between job-related factors, health markers, and well-being consequences. These types of studies would provide a deeper understanding of how workplace variables impact health outcomes over time. Additionally, evaluating the effectiveness of workplace interventions, whether they target PA improvement, dietary habits, or other health parameters, would contribute to the creation of evidence-based strategies suitable for wider application. Furthermore, broadening the study’s focus to include various occupational sectors would allow for more extensive comparisons and a comprehensive understanding of how health indicators vary across different work environments.

## 5. Conclusions

Tyrolean bank employees generally showed an optimal state of health with regard to BMI, PA behaviors (assessed by the WHO GPA Questionnaire), and micronutrient status (evaluated through whole-blood analysis). When analyzing the data to compare low, moderate, and high PA levels, no significant differences were observed between the PA groups for sociodemographic factors, diet type, homocysteine and CoQ10 markers, as well as most vitamins and minerals. However, a significant increase in vitamin D levels was detected as PA levels advanced, highlighting the necessity for targeted interventions to address the increased prevalence of vitamin D deficiency, particularly among individuals with low PA. The present findings provide preliminary evidence concerning the association between PA behaviors, micronutrient status, and health, shedding light on potential implications in the field of occupational health and lifestyle, particularly in the development of individualized approaches (based on PA level) aimed at enhancing the health status of bank employees.

## Figures and Tables

**Figure 1 nutrients-15-04884-f001:**
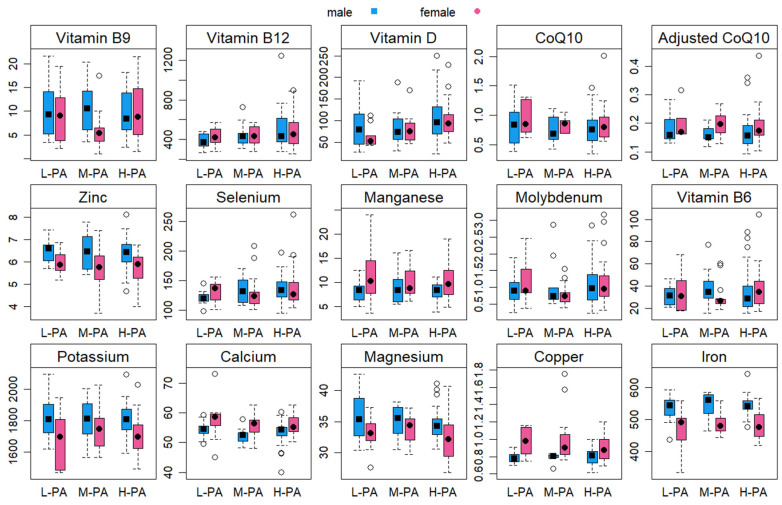
Box plots (with 95% confidence intervals) illustrating the differences in blood variables between participants with low physical activity (L-PA), moderate physical activity (M-PA), and high physical activity (H-PA).

**Figure 2 nutrients-15-04884-f002:**
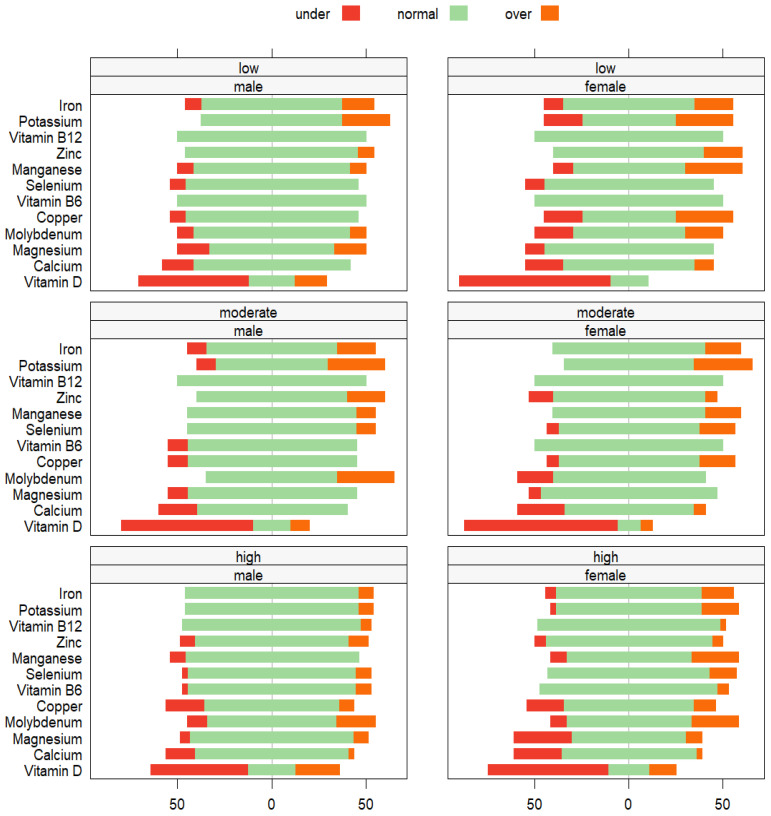
Likert plots displaying the extent of variation of micronutrient values from the sex-specific reference norms for German-speaking populations [39]. Data are categorized based on three PA levels: low, moderate, and high.

**Table 1 nutrients-15-04884-t001:** Comparison of participants’ data across the study groups (based on PA level). Data are presented as percentage, median (with range), or mean ± standard deviation.

		Total (*n* = 123)	Low PA (*n* = 22)	Moderate PA (*n* = 26)	High PA (*n* = 75)	Statistics and *p*-Values
Sex	male (*n* = 61)	50%	55%	38%	52%	χ^2^_(2)_ = 1.68, *p* = 0.432
	female (*n* = 62)	50%	45%	62%	48%	
Age (y)		43 (20–65)	52 (23–62)	45 (20–65)	41 (20–65)	F_(2, 120)_ = 1.68, *p* = 0.191
Body Weight (kg)		71 (43–114)	76 (53–95)	75 (45–114)	69 (43–114)	F_(2, 120)_ = 4.16, *p* = 0.018
Height (cm)		173 (154–190)	172 (163–190)	170 (157–186)	175 (154–190)	F_(2, 120)_ = 0.80, *p* = 0.452
BMI (kg/m^2^)		23.6 (16.9–40.4)	25.1 (19.6–34.1)	24.9 (18.0–40.4)	22.7 (16.9–36.0)	F_(2, 120)_ = 7.63, *p* = 0.001
BMI Levels	<18.5	6%	-	4%	8%	χ^2^_(6)_ = 12.89, *p* = 0.045
	18.5–24.9	61%	50%	46%	69%	
	25.0–29.9	28%	41%	38%	21%	
	≥30.0	5%	9%	12%	1%	
Diet Type	mixed	93%	100%	92%	91%	χ^2^_(2)_ = 2.19, *p* = 0.334
	vegetarian/vegan	7%	-	8%	9%	
Sedentary Time (h/week)		42.5 ± 23.3	53.4 ± 25.9	42.4 ± 20.3	39.3 ± 22.8	F_(2, 120)_ = 3.15, *p* = 0.047
Homocysteine (µmol/L)		10.57 ± 4.30	11.05 ± 2.84	11.80 ± 6.46	10.01 ± 3.63	F_(2, 120)_ = 2.59, *p* = 0.079
Homocysteine Levels	<10	52%	36%	46%	59%	χ^2^_(4)_ = 5.21, *p* = 0.266
	10–15	39%	55%	38%	35%	
	>15	9%	9%	15%	7%	
CoQ10 * (mg/L)		0.82 ± 0.28	0.89 ± 0.34	0.79 ± 0.20	0.81 ± 0.29	F_(2, 79)_ = 0.35, *p* = 0.703
Adjusted CoQ10 * (µmol/mmol Chol)		0.18 ± 0.06	0.19 ± 0.06	0.17 ± 0.04	0.18 ± 0.06	F_(2, 79)_ = 0.44, *p* = 0.644

* *n* = 82. BMI: body mass index; PA: physical activity; CoQ10: coenzyme Q10.

**Table 2 nutrients-15-04884-t002:** Comparison of micronutrient status of female participants based on three PA levels. Data are presented as mean ± standard deviation.

	Females (*n* = 62)				Statistics and *p*-Values
	Total	Low PA	Moderate PA	High PA	
Potassium (mg/L)	1687 ± 234	1684 ± 180	1751 ± 134	1660 ± 277	F_(2, 59)_ = 0.85, *p* = 0.433
Calcium (mg/L)	56.3 ± 4.1	57.7 ± 7.2	55.8 ± 3.4	56.1 ± 3.2	F_(2, 59)_ = 0.71, *p* = 0.497
Magnesium (mg/L)	32.8 ± 3.0	33.0 ± 2.7	33.8 ± 2.4	32.3 ± 3.3	F_(2, 59)_ = 1.89, *p* = 0.161
Copper (mg/L)	0.93 ± 0.20	0.97 ± 0.16	1.01 ± 0.28	0.89 ± 0.15	F_(2, 59)_ = 1.24, *p* = 0.297
Iron (mg/L)	482 ± 43	473 ± 63	488 ± 37	481 ± 41	F_(2, 59)_ = 0.16, *p* = 0.855
Zinc (mg/L)	5.80 ± 0.69	5.97 ± 0.56	5.72 ± 0.91	5.78 ± 0.63	F_(2, 59)_ = 0.25, *p* = 0.778
Selenium (µg/L)	138 ± 42	131 ± 19	145 ± 66	137 ± 32	F_(2, 59)_ = 0.21, *p* = 0.813
Manganese (µg/L)	10.33 ± 3.70	11.39 ± 5.61	9.98 ± 3.12	10.19 ± 3.34	F_(2, 59)_ = 0.22, *p* = 0.806
Molybdenum (µg/L)	1.12 ± 0.83	1.15 ± 0.69	0.79 ± 0.31	1.25 ± 0.98	F_(2, 59)_ = 2.93, *p* = 0.061
Vitamin B_6_ (µg/L)	38.2 ± 30.8	33.4 ± 16.5	29.9 ± 12.1	43.3 ± 38.1	F_(2, 59)_ = 1.55, *p* = 0.221
Vitamin B_9_ (ng/mL)	8.88 ± 5.70	8.74 ± 5.66	5.83 ± 3.87	10.23 ± 5.98	F_(2, 57)_ = 2.98, *p* = 0.059
Vitamin B_12_ (pg/mL)	579 ± 877	438 ± 90	436 ± 97	684 ± 1150	F_(2, 58)_ = 0.45, *p* = 0.641
Vitamin D (nmol/L)	89.9 ± 37.2	63.1 ± 24.3	79.9 ± 30.9	101.9 ± 38.1	F_(2, 59)_ = 7.95, *p* = 0.001

**Table 3 nutrients-15-04884-t003:** Comparison of micronutrient status of male participants based on three PA levels. Data are presented as mean ± standard deviation.

	Males (*n* = 61)				Statistics and *p*-Values
	Total	Low PA	Moderate PA	High PA	
Potassium (mg/L)	1811 ± 113	1827 ± 144	1811 ± 136	1806 ± 98	F_(2, 58)_ = 0.01, *p* = 0.994
Calcium (mg/L)	53.5 ± 3.4	54.5 ± 2.9	52.3 ± 2.8	53.4 ± 3.7	F_(2, 58)_ = 1.97, *p* = 0.148
Magnesium (mg/L)	34.9 ± 2.8	35.8 ± 3.9	35.2 ± 2.5	34.5 ± 2.4	F_(2, 58)_ = 0.68, *p* = 0.511
Copper (mg/L)	0.79 ± 0.09	0.78 ± 0.06	0.79 ± 0.05	0.79 ± 0.10	F_(2, 58)_ = 0.13, *p* = 0.880
Iron (mg/L)	543 ± 34	536 ± 44	546 ± 41	544 ± 28	F_(2, 58)_ = 0.38, *p* = 0.685
Zinc (mg/L)	6.43 ± 0.68	6.52 ± 0.52	6.45 ± 0.81	6.41 ± 0.70	F_(2, 58)_ = 0.11, *p* = 0.894
Selenium (µg/L)	136 ± 35	121 ± 12	134 ± 23	142 ± 41	F_(2, 58)_ = 3.39, *p* = 0.040
Manganese (µg/L)	8.29 ± 2.13	8.14 ± 2.10	8.83 ± 3.27	8.20 ± 1.81	F_(2, 58)_ = 0.07, *p* = 0.933
Molybdenum (µg/L)	1.60 ± 4.23	0.93 ± 0.42	4.36 ± 10.36	1.10 ± 0.61	F_(2, 58)_ = 0.12, *p* = 0.886
Vitamin B_6_ (µg/L)	36.5 ± 22.3	31.4 ± 8.8	38.5 ± 17.9	37.6 ± 26.1	F_(2, 58)_ = 0.38, *p* = 0.684
Vitamin B_9_ (ng/mL)	10.02 ± 4.92	10.22 ± 6.03	10.90 ± 5.05	9.73 ± 4.62	F_(2, 58)_ = 0.16, *p* = 0.851
Vitamin B_12_ (pg/mL)	491 ± 247	381 ± 74	452 ± 127	535 ± 292	F_(2, 58)_ = 3.20, *p* = 0.048
Vitamin D (nmol/L)	99.8 ± 49.9	89.7 ± 51.2	84.1 ± 44.4	107.0 ± 50.5	F_(2, 58)_ = 1.27, *p* = 0.290

## Data Availability

Data are contained within the article. Personal data are not publicly available due to ethical restrictions.

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
