# Peer review of "The Association of Physical Activity Level with Micronutrient and Health Status of Austrian Bank Employees"

_nutrients, 2023, doi:10.3390/nu15234884_

Round 1
Reviewer 1 Report
Comments and Suggestions for Authors
I have several comments on the manuscript, especially in relation to the methodology of the study.
The authors stated that BMI was calculated based on the data provided by the participants. Therefore, I wonder if they were instructed on how to measure the body weight, at what time, what to follow and what to avoid in order for the body weight to be as accurate as possible? Similarly, in the case of body height.
Did the authors detect and monitor any diseases in the participants that can fundamentally affect the nutritional status also in relation to the monitored micronutrients? Or was the use of medication investigated? Alternatively, the menstrual period in women, etc.?
Did the authors investigate the eating habits of the participants with particular regard to the use of supplements? Their use could fundamentally affect the results.
I would also be interested in the possible influence of the level of education on the results achieved and in what period of the year the study was carried out.
Based on this, I recommend the authors to answer the mentioned questions and comments. Finally, I recommend giving the authors a chance to revise the manuscript after minor or major revision.
Author Response
Dear Reviewer #1,
Dear Editors,
Thank you for your consideration, positive feedback, and the opportunity to revise the manuscript “nutrients-2709410” entitled “The Association of Physical Activity Level with Micronutrient and Health Status of Austrian Bank Employees”, which has led to a significant improvement of the manuscript.
In response to your valuable comments, we provided some evidence along with detailed explanations that directly address each comment. We hope our responses below and revisions throughout the manuscript will address the existing concerns of the reviewer and editors. Changes in the manuscript have been highlighted via “Track Changes”.
Kind Regards,
Markus Schauer, on behalf of the team of authors
Review Report 1
-----------------------------------------------------------------------------------------------------------------------------
Comments mentioned by Reviewer #1 and our responses to each comment (in red)
-----------------------------------------------------------------------------------------------------------------------------
I have several comments on the manuscript, especially in relation to the methodology of the study.
Answer: Thank you for reviewing and providing valuable feedback.
The authors stated that BMI was calculated based on the data provided by the participants. Therefore, I wonder if they were instructed on how to measure the body weight, at what time, what to follow and what to avoid in order for the body weight to be as accurate as possible? Similarly, in the case of body height.
Answer: Thank you for expressing your concern. In the present study, detailed instructions were provided for participants on how to report precise values, and particularly, we emphasized that accurate weight measurement requires wearing only underwear, empty digestive tracts, and no exercise prior to the measurement; and accurate height measurement should be conducted without shoes, ensuring that the four body parts (heels, buttocks, shoulders, and the back of the head) touch the wall. However, in line with the reviewer’s concern and the importance of the issue, we believe further explanations are necessary to include in both the Methods and Limitations sections.
Action taken: The relevant explanations have been added to the part 2.2 (lines 157-159) as well as to the paragraph addressing the study limitations (lines 440-443).
Did the authors detect and monitor any diseases in the participants that can fundamentally affect the nutritional status also in relation to the monitored micronutrients? Or was the use of medication investigated? Alternatively, the menstrual period in women, etc.?
Answer: We appreciate your concern and thank you for your valuable comment. The questionnaire encompassed various questions regarding participants' diseases and medication use. Notably, only one participant identified suffering from a severe disease (stomach cancer) which profoundly influenced nutritional status and was subsequently excluded from the final analysis. We understand that an in-depth exploration of diseases, medication use, and physiological variations is important for a better understanding nutritional status; however, we did not exclude participants with non-severe conditions (such as mild hyperglycemia or hyperlipidemia) or those using associated medications. This decision was made to ensure the study encompassed a diverse population, reflecting realistic health conditions (not just investigating healthy individuals only), as we believe that the inclusion of individuals with various health profiles contribute to the generalizability and applicability of the study findings.
Action taken: The relevant explanations have been added to the part 2.1 (lines 139-141).
Did the authors investigate the eating habits of the participants with particular regard to the use of supplements? Their use could fundamentally affect the results.
Answer: Thank you for expressing your concern. The examination of participants’ eating habits, particularly in relation to supplement use, was not explicitly addressed in the present study. While we understand the significance of this aspect and acknowledge it as a potential limitation, it is noteworthy that the present study was designed to offer preliminary evidence regarding the association between PA behaviors and micronutrient status, aiming to shed light on potential implications in the fields of occupational health and lifestyle. Therefore, a more comprehensive analysis of this topic, considering all confounding variables (including dietary confounders, especially supplement use), is reserved for our future research efforts. Thank you for your kind understanding.
Action taken: We strengthened the limitation list by specifying “eating habits” and “supplement use” as two potential confounding factors (line 450).
I would also be interested in the possible influence of the level of education on the results achieved and in what period of the year the study was carried out.
Answer: Thank you for your comment. The study did not explicitly explore the education levels of participants as the target population (i.e., Austrian bank employees) typically exhibits a limited variation in education. As a prerequisite in Austria to apply for employment in bank sector, it is expected to provide a higher education qualification that accomplishes with A-Level academic graduation with an economic orientation and focus, and thus, candidates are generally well-educated. Regarding the second part of the comment, it is notable that we collected the data during December 2019 and January 2020.
Action taken: We included the time of data collection in part 2.1 (line 129).
Based on this, I recommend the authors to answer the mentioned questions and comments. Finally, I recommend giving the authors a chance to revise the manuscript after minor or major revision.
Answer: Thank you for your valuable time and consideration. We hope our clarifications and revisions have addressed your concerns.
Reviewer 2 Report
Comments and Suggestions for Authors
Dear Authors,
The article submitted for review concerns the important issue of differences in the biological state, and specifically in the health state, between adult, professionally active men and women who differ significantly in the level of physical activity and the number of sedentary hours. Although the research group is relatively small, the study seems interesting. My comments concern mainly the details of the research methodology and theoretical chapters.
Comments and suggestions for authors:
1. Please explain why the authors did not perform anthropometric measurements themselves in accordance with the procedures and anthropometric research protocol, since they had direct contact with the subjects? Did the respondents receive detailed instructions on how to perform anthropometric measurements on their own? Data on height and weight used to assess BMI may be subject to at least two errors, and therefore the assessment of BMI and, consequently, the occurrence of overweight, obesity and underweight may be very incorrect. Differences between subjectively declared and objective anthropometric measures may be highly significant and differentiated between genders.
I recommend: https://pubmed.ncbi.nlm.nih.gov/26133040/
I draw attention to this problem because the authors did not refer to this tool neither in the methodological chapter nor in the Limitation study. Please make appropriate additions to the manuscript.
2. Serious note - Cut-points for BMI categories should be precisely corrected throughout the manuscript - in accordance with WHO. In most places in the manuscript they are given imprecisely, i.e. incorrectly (e.g. use right- or left-closed range intervals, respectively).
3. In my opinion, providing average values of body weight, body height, BMI or even age for women and men is absolutely inappropriate. Note the enormous diversity of anthropometric characteristics in Homo sapiens, which is a species with a moderate amount of sexual dimorphism. Please provide the values of these features separately for men and women - including age, which is extremely wide (20-65 years old!).
4. Given the relatively small number of men and women surveyed (approximately 60 people each), the age range of 20-65 years constitutes a serious limitation of the study. In the study group, the biological condition of a young man in a progressive period of development and of a postmenopausal woman or man with severely advanced andropausal changes can be compared. According to the methodology, narrowing the age of the study group or using age standardization should be considered. Please consider and make appropriate additions, also in the Limiatation study.
5. Another important limitation of the study is that only professionally active people were examined (https://pubmed.ncbi.nlm.nih.gov/20627735/)
6. Please complete the section regarding the material - when the research was performed, what were the exclusion and inclusion criteria for the study, whether the length of service in the bank was known, whether there was information about the position and professional position of the surveyed employees, how many hours they usually work and whether they have a break from work ( how long), whether it is shift work, etc.
7. Please provide a detailed description of who withdrew from participating in the study - and whether any parameters of these people are known. Ultimately, the results obtained by these officials indicate their surprisingly good biological condition.
8. Please describe in detail what is presented in tables 2 and 3. Are these SD or SE values?
9. Please refer to the differences in vitamin concentration. D between subjects with different PA in the Discussion section. Is it known whether the declared physical activity was indoors or outdoors (do the authors know the forms of physical activity of the respondents?)
10. In my opinion, the order of presentation of results in figures and tables must be the same - please standardize it.
11. In my opinion, in each study concerning a specific narrow professional group, in the theoretical sections, especially in the discussion, the issue of the relationship between the current biological state and 1. the length of years of work (work experience) and 2. the directional selection for the profession in terms of biology should be addressed.
I recommend: https://pubmed.ncbi.nlm.nih.gov/27690723/
Kind regards,
reviewer
Author Response
Dear Reviewer #2,
Dear Editors,
Thank you for your consideration, positive feedback, and the opportunity to revise the manuscript “nutrients-2709410” entitled “The Association of Physical Activity Level with Micronutrient and Health Status of Austrian Bank Employees”, which has led to a significant improvement of the manuscript.
In response to your valuable comments, we provided some evidence along with detailed explanations that directly address each comment. We hope our responses below and revisions throughout the manuscript will address the existing concerns of the reviewer and editors. Changes in the manuscript have been highlighted via “Track Changes”.
Kind Regards,
Markus Schauer, on behalf of the team of authors
Review Report 2
-----------------------------------------------------------------------------------------------------------------------------
Comments mentioned by Reviewer #2 and our responses to each comment (in red)
-----------------------------------------------------------------------------------------------------------------------------
Dear Authors,
The article submitted for review concerns the important issue of differences in the biological state, and specifically in the health state, between adult, professionally active men and women who differ significantly in the level of physical activity and the number of sedentary hours. Although the research group is relatively small, the study seems interesting. My comments concern mainly the details of the research methodology and theoretical chapters.
Answer: Your feedback is highly valued and we are grateful for your considerate comments.
Comments and suggestions for authors:
- Please explain why the authors did not perform anthropometric measurements themselves in accordance with the procedures and anthropometric research protocol, since they had direct contact with the subjects? Did the respondents receive detailed instructions on how to perform anthropometric measurements on their own? Data on height and weight used to assess BMI may be subject to at least two errors, and therefore the assessment of BMI and, consequently, the occurrence of overweight, obesity and underweight may be very incorrect. Differences between subjectively declared and objective anthropometric measures may be highly significant and differentiated between genders. I recommend: https://pubmed.ncbi.nlm.nih.gov/26133040/
I draw attention to this problem because the authors did not refer to this tool neither in the methodological chapter nor in the Limitation study. Please make appropriate additions to the manuscript.
Answer: We appreciate your valuable and insightful comment regarding the use of self-reported height and weight data in calculating BMI. Although we did not perform direct anthropometric measurements ourselves (due to some technical restrictions), we provided detailed instructions for participants on reporting precise values (i.e., emphasizing that accurate weight measurement requires wearing only underwear, empty digestive tracts, no exercise prior to the measurement; and accurate height measurement should be conducted without shoes, ensuring that the four body parts (heels, buttocks, shoulders, and the back of the head) touching the wall. However, in line with the reviewer’s concern and the importance of the issue, we believe clarifying it in the section Methods and also listing it as a study limitation are necessary to improve transparency and completeness of the study.
Action taken: The relevant explanations have been added to the part 2.2 (lines 157-159) as well as to the paragraph addressing the study limitations (lines 440-443).
- Serious note - Cut-points for BMI categories should be precisely corrected throughout the manuscript - in accordance with WHO. In most places in the manuscript they are given imprecisely, i.e. incorrectly (e.g. use right- or left-closed range intervals, respectively).
Answer: Thank you for bringing attention to this important point.
Action taken: We had a careful check throughout the manuscript and made revisions to align the presentation of BMI cut-off points with the guidelines outlined by the WHO (lines 162-163 and 305, and Table 1).
- In my opinion, providing average values of body weight, body height, BMI or even age for women and men is absolutely inappropriate. Note the enormous diversity of anthropometric characteristics in Homo sapiens, which is a species with a moderate amount of sexual dimorphism. Please provide the values of these features separately for men and women - including age, which is extremely wide (20-65 years old!).
Answer: Thank you for emphasizing the significance of sex-based differences in anthropometric characteristics. While we acknowledge your concern, it’s important to note that our study involved a one-step sample differentiation by categorizing participants into PA groups. Statistically, it was not feasible to further differentiate the study groups into smaller sex-based subgroups. While we agree that this issue should be presented as a limitation, it is also noteworthy that the primary objective of the present study was to provide initial evidence regarding the association between PA behaviors and micronutrient status, aiming to shed light on potential implications in the fields of occupational health and lifestyle. Hence, a more comprehensive and differentiated analysis of this topic is reserved for future research efforts, including ours, as we are currently preparing two separate research reports/papers from this project, wherein we separately analyze age-based and sex-based differences in micronutrients and health status of the target population. Thank you for your understanding.
Action taken: The relevant explanations have been added as a study limitation (lines 451-453).
- Given the relatively small number of men and women surveyed (approximately 60 people each), the age range of 20-65 years constitutes a serious limitation of the study. In the study group, the biological condition of a young man in a progressive period of development and of a postmenopausal woman or man with severely advanced andropausal changes can be compared. According to the methodology, narrowing the age of the study group or using age standardization should be considered. Please consider and make appropriate additions, also in the Limiatation study.
Answer: Thank you for your valuable and insightful comment. In alignment with the points mentioned in the previous comment, we fully acknowledge that the wide age range (while scientifically representative of the “adulthood” life period) may introduce potential biases into the findings. On the other hand, However, the wide age range in our sample precisely reflects the complete distribution of the target population, as bank employees are employed from the age of 20 and continue to work until 65 (which is the retirement age in Austria). But regarding this specific study, while no age-based difference was observed between the study groups, the sample size limitations across the study groups prevented us from further differentiating the PA groups (as target variable and major focus of the present paper) into age-based subgroups. However, considering the scope of project, we want to assure the esteemed reviewer that the significance of age-based comparisons with full details will soon be addressed in a separate research report. We appreciate your kind understanding.
Action taken: As mentioned earlier, the relevant explanations have been added as a study limitation (lines 451-453).
- Another important limitation of the study is that only professionally active people were examined (https://pubmed. ncbi.nlm.nih.gov/20627735/)
Answer: Thank you for your comment and for providing a link for further information. We acknowledge the need to include additional explanations in the section Limitations about the generalizability of our findings, given the fact that the target population was a specific professional group.
Action taken: The relevant explanations have been added to the limitation statement associated with the target population (lines 448-449), and the suggested reference has also been added to the study references to support the statement (ref: 93).
- Please complete the section regarding the material - when the research was performed, what were the exclusion and inclusion criteria for the study, whether the length of service in the bank was known, whether there was information about the position and professional position of the surveyed employees, how many hours they usually work and whether they have a break from work (how long), whether it is shift work, etc.
Answer: We greatly appreciate your detailed and valuable suggestions for improving the Methods section, which we agree with. While many of this information is available to be added, a few details, such as their exact positions and break times during work, were unknown. It is noteworthy that the only specific exclusion criterion was “suffering from a severe health condition”, and consequently, we excluded one participant who disclosed having stomach cancer.
Action taken: We strengthened different parts of the section Methods incorporating additional information about the participants and the study (lines 128, 129, 136-137, 139-141).
- Please provide a detailed description of who withdrew from participating in the study - and whether any parameters of these people are known. Ultimately, the results obtained by these officials indicate their surprisingly good biological condition.
Answer: Thank you for your comment. Out of the initial sample of 280 bank employees who expressed interest in participating, 123 participants successfully completed all tests and measurements. With the exception of one participant with stomach cancer who was intentionally excluded, the primary reason for the relatively high drop-out rate was the inability to provide complete and reliable data. Therefore, due to the cross-sectional nature of this study, data from those not included in the final sample were not analyzed, and consequently, no additional information about their sociodemographic factors is available for reporting.
Action taken: The relevant explanations regarding the exclusion criterion have been added to Section 2.1 (lines 139-141).
- Please describe in detail what is presented in tables 2 and 3. Are these SD or SE values?
Answer: Thank you for bringing this point to our attention.
Action taken: The relevant information has been added to the headings of Table 2 and Table 3 (lines 277-278, 280-281), as well as Table 1 (lines 260-261).
- Please refer to the differences in vitamin concentration. D between subjects with different PA in the Discussion section. Is it known whether the declared physical activity was indoors or outdoors (do the authors know the forms of physical activity of the respondents?)
Answer: We appreciate your attention to this important point. In the present study we used the GPAQ (developed by the WHO) to assess PA parameters (particularly PA level and sedentary behaviors). The WHO-GPAQ assesses PA across three domains: work, travel, and recreational activities using different questions about the intensity, frequency, and duration of activities in these domains. The WHO-GPAQ, however, does not explicitly specify whether the activity occurred indoors or outdoors. Nevertheless, in line with your valuable comment, we believe that the Discussion section can be further strengthened with an explanation regarding the increased level of vitamin D among female participants.
Action taken: A explanatory statement, supported with a new reference (ref: 75), has been added to the associated contents (lines 386-389).
- In my opinion, the order of presentation of results in figures and tables must be the same - please standardize it.
Answer: Thank you for bringing this point to our attention.
Action taken: To enhance the flow of content, we relocated Figure 1 to follow Tables 2 and 3 (lines 283-286), while also revising the corresponding explanations in the text (lines 257-258, 274-276).
- In my opinion, in each study concerning a specific narrow professional group, in the theoretical sections, especially in the discussion, the issue of the relationship between the current biological state and 1. the length o years of work (work experience) and 2. the directional selection for the profession in terms of biology should be addressed. I recommend: https://pubmed.ncbi.nlm.nih.gov/27690723/
Answer: Thank you for your valuable comment and for providing a link for further information.
Action taken: We strengthened the Section Discussion with additional explanations about the importance of the long-term impact of the working environment as well as biological predispositions (lines 434-437), and the suggested reference has also been added to support the statement (ref: 92).
Round 2
Reviewer 2 Report
Comments and Suggestions for Authors
Dear Authors,
Thank you very much for the opportunity to review this article for the second time. The authors responded exceptionally carefully to my questions and suggestions and introduced appropriate corrections to the manuscript.
Thank you for agreeing with almost all of your comments. The promised Research Report will certainly be very interesting. I accept this version of the manuscript. Good luck.
Kind regards,
reviewer